# Absolute and Relative Strength, Power and Physiological Characteristics of Indian Junior National-Level Judokas

**DOI:** 10.3390/sports8020014

**Published:** 2020-01-28

**Authors:** Dale M. Harris, Kristina Kendall, G. Gregory Haff, Christopher Latella

**Affiliations:** 1First Year College, Footscray Park Campus, Victoria University, Melbourne 3011, Australia; 2High Performance Department, Inspire Institute of Sport, Vidyanagar, Bellary 583275, India; 3Centre for Exercise and Sports Science Research, School of Medical and Health Sciences, Edith Cowan University, Joondalup 6027, Australia; k.kendall@ecu.edu.au (K.K.); g.haff@ecu.edu.au (G.G.H.); c.latella@ecu.edu.au (C.L.); 4Directorate of Sports, Exercise and Physiotherapy, University of Salford, Greater Manchester M5 4WT, UK; 5Neurophysiology Research Laboratory, Edith Cowan University, Joondalup 6027, Australia

**Keywords:** judo, combat sport, strength, power, martial arts, performance

## Abstract

The physical qualities that underpin successful junior judokas requires continuing investigation. We investigated the physical and physiological characteristics of junior national level judokas. We tested 25 (15 male, 10 female) Indian judokas for absolute and relative strength (back-squat and bench-press one-repetition maximum (1RM) as well as isometric handgrip), aerobic (RAMP test) and lower-body anaerobic power (Wingate 6-s sprint and countermovement jump), change-of-direction (5-0-5 test) and speed (30 m sprint). Athletes were grouped according to national-level competition placing (gold-medal winners (GM; n = 8), all medal winners (MW; n = 13), non-medallists (NM; n = 12), and NM plus silver and bronze; all others (AO; n = 17)). Stepwise discriminant function analysis determined characteristics likely to predict successful performance. Independent t-tests and effect size (Hedge’s g) analyses were performed between groups. GM demonstrated greater lower-body absolute (20.0%; g = 0.87, p = 0.046) and relative 1RM strength (21.0%; g = 0.87, p = 0.047), and greater lower-body absolute (25.4%; g=1.32, p=0.004) and relative (27.3%; g = 1.27, p = 0.005) anaerobic power compared to AO. Furthermore, anaerobic power can correctly predict 76.5% and 62.5% of AO and GM athletes, respectively. No differences were observed between MW and NM groups. The results suggest the importance of lower-body strength and power for junior national-level judokas and provides information for professionals working with these athletes.

## 1. Introduction

Sport is often characterised by highly competitive and closely fought competition, with small margins often differentiating successful and non-successful performance outcomes [1,2]. At elite and sub-elite (e.g., national) levels of competition, optimal physical preparation can be a key determinant of sporting success. Thus, seemingly small improvements in physical and physiological qualities may translate to a significant advantage for athletes. In combat sports, early research has studied these factors in elite adults [3,4,5,6,7]. However, over the past decade participation in combat sports, including judo, has grown considerably, resulting in increased junior participation and professionalism of competitions. In turn, this has advanced the need for further scientific investigation into the specific physiological and performance requirements of junior national-level judokas. Specifically, the requirements of elite junior judokas may differ to adult counterparts and other combat sports due to biological maturity status and the unique tactical, technical and physical requirements of competition, respectively.

A plethora of research is now available investigating the physical and physiological factors differentiating more (i.e., medal-winning), from less (i.e., non-medallists) successful athletes in various individual and team-based field sports [6,8,9,10,11,12], and even across weight divisions [13]. In combat sports, recent evidence has demonstrated that physical performance qualities contribute to success in Brazilian jiujitsu [14], taekwondo [10] and mixed martial arts [15]. In mixed martial arts for example, James et al. [16] established that lower-body performance characteristics, particularly dynamic strength, were significantly greater in more successful compared to less successful athletes. To date, although there are a substantial number of studies which have examined the physical performance qualities underpinning successful competition performance in adult judokas [7], less evidence exists in juniors. This is particularly important, as many junior and sub-junior national level competition results decide who will take part in national training camps to prepare for international competition; typically, national competition gold-medal winners will be prioritised a place at the national training camp in India, for example.

According to previous work, there are several physical performance qualities that may be particularly important for judokas to develop [12]. For example, greater upper-body anaerobic peak and mean power, and specific technical skill performance (achieving more throws in the allotted time) in the special judo fitness test (SJFT), were noted for more successful compared to less successful adult judokas. Additionally, Barbado et al. [17] reported greater trunk extensor strength for more successful compared with less successful adult judokas. However, despite the importance of trunk strength for the optimal execution of throwing techniques (e.g., Uchi mata), evidence regarding upper-body strength and power remains limited when examining this sport [18], particularly with regards to junior judokas. Akin to the evidence in adult judokas [7], it is likely that greater upper- and lower-body strength and power capabilities of junior judokas may contribute to competition performance outcomes. Additionally, maximal and sustained intermittent handgrip strength also appears to be an important factor in successful junior judo performance [19]. However, further scientific evidence is required to fully explore how upper and lower-body strength, anerobic and aerobic power and speed qualities may contribute to junior judoka performance. 

Thus, the aim of this study was to examine the various physical and physiological characteristics (upper- and lower-body strength and anaerobic/aerobic power and capacity) of junior national-level Indian judokas, and specifically which factors may best predict a gold-medal (GM) winning performance. We hypothesised that successful junior national-level judokas would possess greater upper- and lower-body strength and power qualities. The results of this investigation are intended to provide important information for strength and conditioning professionals regarding the optimal preparation of junior national-level judokas to facilitate successful performance during high-level competition.

## 2. Materials and Methods

### 2.1. Experimental Approach to the Problem

The data for this study was analysed post-hoc in a retrospective case-controlled design. Physical and physiological testing was conducted over three separate days, with at least 24 h between testing sessions (Figure 1). Each day of testing began with a 10-min warm-up on a treadmill at a velocity corresponding to 50%–60% of the estimated VO_2_max, derived from the athlete’s most recent 30–15 intermittent fitness test score [20]. The warm-up also comprised specific upper- and lower-body movement preparation exercises consisting of 10 repetitions of each exercise; banded crab walks, bodyweight deep squats, inch worms with rotations, walking lunges with rotations, A-skips and drop-landings (performed on a plyometric box). During all tests, the athlete’s best result was recorded and used for subsequent analysis.

### 2.2. Subjects

A total of 15 male and 10 female light- to middle-weight Indian judokas were included in this retrospective case-controlled study. Given the limited number of athlete’s in each category (i.e., gold medal (GM), medal winning (MW), non-medallists (NM) and all others (AO)), sex differences were not explored. The mean age, height and bodyweight of the athletes in each group and for each sex are presented in Appendix A. All athletes had >4 years of competitive national-level experience, of which national competitions followed standard international Judo federation rules and regulations, and had >2 years of resistance training experience and were familiar with all the tests conducted. Moreover, each athlete had been practicing Judo for at least >5 years, and many were preparing for national competitions to be selected for national training camps in order to compete at upcoming international competitions (e.g., Youth Olympic Games). Athletes were not in a weight cutting phase during the period of testing. Additionally, all athletes were free from injury and able to participate in full training at the time of testing. All testing was conducted at the Inspire Institute of Sport, India, in accordance with Edith Cowan University Human Research Ethics Committee (ethics approval number: 21984).

### 2.3. Testing Day One

The first test consisted of an isometric handgrip (BMS Hydraulic Hand Dynamometer, Bharat Medical Systems, Chennai, India) assessment (left and right), which has a high test-retest reliability (Intraclass correlation coefficient (ICC) = 0.87) [21]. Athletes were instructed to stand with their shoulder adducted, elbow flexed at 90° and their forearm and wrist in a neutral position. The grip of the dynamometer was orientated to each athlete’s metacarpophalangeal joint and athletes were instructed to maintain a maximal 5 s isometric contraction during each attempt. Each athlete performed three maximum trials on each hand, in an alternating fashion, with one-minute recovery between attempts. The peak value from each hand was recorded and used for further analysis. Next, lower-body power was assessed via a countermovement vertical jump (CMJ) performed on a portable jump mat (Probotics, Huntsville, AL, USA) using the protocol previously described by Markovic et al. [22] which has been reported as highly reliable (ICC = 0.98). Each CMJ attempt required athletes to place their hands on their hips and using a self-selected countermovement, jump vertically as high as possible and land in the same position on the mat. Each athlete performed three CMJ trials with one-minute recovery between each attempt. Vertical jump height was determined by converting flight-time into jump height with the following conversion equation:
$$height\;\left(m\right)\;=\frac{t_{f}\times t_{f}\times \;g}{8}$$
where *t_f_* is flight time in seconds and *g* is the acceleration of gravity (9.81 m·s^−2^). The CMJ height score (in metres) was then converted into peak power (measured in Watts) by means of the Sayers equation for the squat jump [23]:
Peak Power (W)= (60.7) × (jump height (cm)) + 45.3 × (body mass (kg)) − 2055

Following this, athletes were tested for a one-repetition maximal (1RM) back squat performed with a competition standard Olympic bar (20 kg) and weight plates (Fitness World, U.P., IND). Each athlete performed 10 repetitions at a low intensity before gradually increasing the load in 5%–10% increments until they could not complete a full repetition. The number of sets before reaching the 1RM was four to six, and a three-min recovery period was granted between each set. Following a five-minute break, athletes then performed a 1RM bench press using the same protocol used for the 1RM squat. Maximal strength testing (i.e., 1RM back squat and bench press) have consistently demonstrated excellent reliability when performed with standardised protocols (ICC = 0.99 both tests) [24].

### 2.4. Testing Day Two

All athletes performed three attempts of a flying 30 m sprint on a permeable latex-bound synthetic indoor running surface, with a three-minute recovery period provided between each sprint. Athletes were instructed to begin from a stationary position, and initiate each sprint upon the verbal command of “Go”. Athletes, then performed a total of six 5-0-5 change of direction sprints in a randomised order, with three attempts performed with both the left and right foot and a two-min recovery period between each attempt. Athletes were instructed to begin from a stationary start position, and initiate each sprint upon the verbal command of “Go”. Both the flying sprint times and the 5-0-5 change of direction sprint times were recorded using Brower timing gates (Brower Timing Systems, Draper, UT, USA), which have been consistently shown to provide reliable sprint (Flying 30 m sprint ICC = 0.99) [25] and change of direction (5-0-5 ICC = 0.96) times [26]. 

### 2.5. Testing Day Three

Lower-body peak anaerobic power was assessed via a six-second maximal bike sprint test (Wattbike Pro, Nottingham, UK) using the protocol described by Herbert et al. [27]. Body mass was measured to the nearest 0.1 kg using a free weight scale (HealthSense PS 126 Ultra-Lite Personal Scale, Bright Health Care, Bangalore, India). Prior to conducting the test, the bike was calibrated according to manufacturers’ guidelines. Saddle height, seat-post angle (relative to crank fulcrum), handlebar height and distance to saddle were individualised for each athlete. Saddle height was adjusted relative to the crank position and the foot was secured to a pedal with clips. Athletes began the test in a seated stationary position with their dominant leg initiating the first down-stroke. The air brake resistance was set to level 10, and magnetic resistance set to level one as per standard testing protocols. Before initiation, the test was preceded by a 5-min warm-up at resistance level eight, corresponding to a Borg derived rating of perceived exertion (RPE) 11–13 (light to somewhat hard) and incorporated two acceleration phases of approximately 3 seconds commencing at the 90 and 180 s mark. A five-min recovery period followed the warm-up. The test was then initiated after a 5-s countdown followed by a firm verbal command of “Go”. Strong verbal encouragement was maintained throughout the attempt, and completed with a final verbal command of “Stop”. Following this, all athletes performed a five-minute cool-down with the air brake resistance level eight. The peak anaerobic power output derived from the test was manually recorded in an excel data spreadsheet and used for analysis. 

Athletes were given a 10-min recovery period between the anaerobic peak-power test and the aerobic power test. The aerobic power (VO_2max_) was predicted using an identical set-up to the anaerobic peak-power test conducted on the Wattbike Pro (Wattbike, West Bridgford, Nottingham, UK). However, the test used in this instance was the maximal RAMP test. Athletes completed a 5-min warm-up at the starting power for the test (120 W for males, 80 W for females). After the warm-up, the required work rate increased by 20 W each minute. Athletes maintained a cadence of between 90–100 revolutions per minute (RPM) and continued cycling until volitional exhaustion. Strong verbal encouragement was provided by the researchers throughout the test. The VO_2max_ scores were estimated from the peak power outputs achieved during the RAMP test using the following regression equation [28]:
$$VO_{2max}\;\;\left(mL\cdot kg^{-1}\cdot m^{-1}\right)\;=\;\left(10.97\;\times \;peak\;power\;output\;\left(W/kg\right)\right)\;+\;2.598$$

### 2.6. Statistical Analyses

Athletes were grouped according to their placing from their most recent national-level competition and were classified as: gold-medal winners (GM; *n* = 8), gold-, silver- and bronze-medal winners (MW; *n* = 13), non-medallists (NM; *n* = 12), and combined silver-, bronze- and non-medallist athletes; which were termed all others (AO; *n* = 17). This method of categorising athletes based on their competition performances to delineate either physiological or specific technical differences among gold medal winners compared with non-medallists or other podium finishers has been previously conducted [29,30,31,32]. A flowchart of the group formations is illustrated in Figure 2. All data is presented as mean ± standard deviation. Between group differences are presented as a percentage (%) using the following formula: % difference = ((Mean Group 1−Mean Group 2)/Average Group 1&2)×100. All data was screened for normality assumption using a Shapiro–Wilk test, and independent *t*-tests were used to compare the means of the two groups; 1. GM versus AO, and 2. MW versus NM. Significance was set at p<0.05. Additionally, standardised effect size analysis (Hedges’s *g*) and 95% confidence intervals (95% CI) were calculated between the groups [1]. The magnitude of each effect size was quantified as follows; trivial <0.20, small 0.20–0.49, moderate 0.50–0.79, or large >0.8 [1]. Due to the sample size and number of independent variables a discriminant function analysis was conducted to determine which, if any, outcome variable(s) best predict a GM performance. The stepwise method was performed to remove independent variables not considered significant in the model and the F-value set at 3.84 for entry and 2.71 for removal. The results of discriminant value analysis are reported as the p-value and percentage that the independent variable can correctly predict a GM winning or NM athlete. All analyses were conducted using Microsoft Excel version 2016 (Microsoft Corporation, Redmond, WA, USA) and SPSS v.25 (IBM Corporation, Armonk, NY, USA).

## 3. Results

### 3.1. Anaerobic Power

A significant large effect was noted for absolute (25.4%; g = 1.32 (95% CI = 0.40, 2.24), p = 0.004) and relative (27.3%; g = 1.27 (95% CI = 0.36, 2.18), p = 0.005) lower-body anaerobic power for GM compared to AO athletes (Figure 3, Appendix A). Non-significant moderate effects were noted for absolute (17.9%; g = 0.84 (95%CI = 0.02, 1.66) p = 0.091) and relative (16.3%; g = 0.68 (95%CI = −0.12, 1.49), p = 0.091) anaerobic power (Figure 4, Appendix A) for MW compared to NM. No differences were observed between groups for CMJ jump performance. 

### 3.2. Aerobic Power

A non-significant, but moderate negative effect was noted for aerobic power (−7.9%; g = −0.51 (95% CI = −1.36, 0.34), p = 0.233) for GM compared to AO athletes. A non-significant, but moderate negative effect was noted for aerobic power (−6.4%, g = −0.41 (95% CI = −1.20, 0.38), p = 0.302) for MW compared to NM. 

### 3.3. Strength

Significant large effects were noted for absolute (20%; g = 0.87 (95% CI = −0.01, 1.74), p = 0.046) and relative (21%; g = 0.87 (95% CI = 0.00, 1.74), p = 0.047) 1-RM squat strength for GM compared to AO (Figure 5). For the MW group compared to the NM group non-significant moderate effects were noted for absolute (17.0%; g = 0.71 (95% CI = −0.10, 1.52), p = 0.079) and relative (16.2%; g = 0.64 (95% CI = −0.16, 1.45), p = 0.111) 1-RM squat strength, and for absolute 1-RM bench press (12.1%; g = 0.49 (95% CI = −0.30, 1.29), p = 0.217). Moreover, non-significant moderate effects were noted for handgrip strength (5.2%; g = 0.61 (95% CI = −0.19, 1.42), p = 0.126) for MW compared to NM. Relative handgrip scores demonstrated small, non-significant effects between GM and AO groups (1.0%; g = 0.08 (95% CI = −0.76, 0.92) p = 0.852) and between MW and NM groups (3.7%; g = 0.30 (95% CI = −0.49, 1.09), p = 0.467). 

### 3.4. Speed

For the GM group compared to the AO group non-significant moderate effects were noted for 5-0-5 change of direction speed (right side) (−9.1%; g = 0.59 (95% CI = −0.26, 1.45), p = 0.226) and 30 m flying sprint time (−11.9%; g = 0.97 (95% CI = 0.08, 1.85), p = 0.097) (Figure 6).

### 3.5. Power

For the GM group compared to the AO group a non-significant, trivial effect was noted for CMJ power (0.9%; g = 0.05 (95% CI = −0.79, 0.89), p = 0.906). For the MW group compared to the NM group a further non-significant, trivial effect was noted for CMJ power (3.9%, g = 0.17 (95% CI = −0.61, 0.96), p = 0.662). Small or no effects were shown between groups for all other physical performance tests.

### 3.6. Predictors

Discriminant value analysis suggests that only absolute anaerobic power was a significant predictor of GM and AO athletes (p = 0.004) in this cohort. Classification results demonstrate that anaerobic power can correctly predict 76.5% and 62.5% of AO and GM athletes, respectively. 

## 4. Discussion

Although physiological profiling of judokas has increased, less evidence is available in competitive junior national-level judokas. Thus, we examined the physical and physiological characteristics of GM and AO, as well as MW and NM athletes, respectively. Collectively, our results indicate that GM athletes express greater absolute and relative lower-body maximal strength (i.e., squat) and anaerobic peak power (i.e., cycling test) when compared to AO athletes. In addition, moderate effects were noted for relative handgrip strength and maximal upper-body strength (i.e., bench press) for the MW group compared to the NM group. Based upon these findings, upper and lower-body strength and power appear to be important qualities for junior competitive judokas and may, at least in part, contribute to competition performance. Together, our results provide important information for strength and conditioning professionals regarding areas of emphasis for the physical preparation of junior national-level judokas.

Absolute and relative peak strength were significantly higher for GM when compared to AO athletes, and is similar to that observed in other combat sports [16,33]. The importance of the summation of forces in rotational sports (e.g., the expression of lower-body force correlating to improved upper-body performance) has been demonstrated in elite handball [34] and baseball athletes [35,36], although limited research has been conducted in combat sports, particularly in international or national-level judokas. However, Loturco et al. [37] established that lower-body strength (r^2^ = 0.66) and power (r^2^ = 0.77) correlate to punch velocity, and accounted for 65% of the variation in punching acceleration in national-level karate athletes. Although judokas are not required to strike, punching requires a certain amount of torso rotation not unlike the early rotational movements seen in uchi mata techniques. Additionally, lower-body maximal strength and power training can improve spinal musculature strength [38]. Specifically, Barbado et al. [17] demonstrated that international judokas express significantly higher trunk extensor strength compared to their national-level counterparts. As such, it can be reasonably concluded that judokas should possess a high level of lower-body strength and power which may allow for both improved spinal stability and enable them to express greater rotational power during uchi mata techniques. 

There was also a significant difference for lower-body absolute and relative peak cycling anaerobic power between GM and AO athletes. Further, anaerobic power was considered a significant predictor of GM and AO athletes and thus, warrants at least some discussion. For example, Franchini et al. [39] proposed that the phosphagen bioenergetic system is the predominant energy supplier during the SJFT, which is designed to simulate judo competition. The greater phosphagen contribution appears to be a result of the high-intensity efforts performed during the test, and reflective of the intermittent nature of judo competition itself. In particular, the work: rest ratios of judo competition are between 2:1 to 3:1 during competition, with each grappling technique typically lasting 20-30 s alternated with 10 s recovery intervals between high-intensity efforts [40]. Therefore, although a direct relationship between anaerobic power and gold-medal success cannot be established in the present study, the results of the discriminant function analysis suggest that lower-body anaerobic power generating capacity is an important physiological quality underpinning judo performance. Subsequently, we suggest that anaerobic power development should be incorporated into programs aimed at developing and preparing national-level junior Judokas. 

No significant differences were noted for absolute or relative isometric handgrip strength between any group. These results reflect the findings of Franchini et al. [12] and Bonitch-Góngora et al. [19] who also reported small but non-significant differences between medallists and non-medallists in isometric handgrip strength for youth national-level judokas. However, we did find a moderate effect for absolute handgrip between MW and NM groups. Nonetheless, we suggest that isometric handgrip strength is still an important quality for national-level junior judokas, especially when considering the repeated nature of handgrip and throwing techniques in judo competition. Indeed, Bonitch-Góngora et al. [19] established that elite judokas recorded higher relative isometric handgrip strength during all contractions of the sustained intermittent endurance handgrip fitness test compared to non-elite judokas. Thus, in future studies we acknowledge that assessing endurance in conjunction to brief, maximal isometric handgrip strength may provide additional relevant information in this demographic.

Furthermore, our results demonstrated no, or even a negative difference in aerobic power for GM compared to AO, and between MW and NM athletes, respectively. This finding corroborates the results from Franchini et al. [12] who suggested that aerobic power (i.e., VO_2max_) may not be a discriminatory quality in performance outcomes for sub-elite judokas. Though, some caution is required in the interpretation of these results as the predictive VO_2max_ scores are based off cycling performance and may underestimate VO_2max_ by up to 10% compared to running-based (e.g., treadmill) measures [41]. Nevertheless, aerobic power appears to be beneficial for high intensity intermittent combat sports [6,10] and contributes to faster recovery times between competitive judo matches [42]. Therefore, although our results suggest that aerobic power may not contribute to competition performance, we suggest that developing the aerobic system of junior judokas to the values indicated in Franchini et al. [6] may support recovery during longer match durations or between matches. 

Regarding the distribution of males and females, there was a reasonably consistent spread of males and females in each of the groups in our study, respectively. However, we acknowledge that a small sample size makes it difficult to see any statistical differences regarding participant characteristics. Indeed, recent research in the physiological differences between men and women has suggested that, when matched for relative strength, there are no differences in strength and power between sexes [43,44]. As such, although we did not match groups for strength given the nature of our retrospective case-controlled study design, we did establish a significant result for relative (to bodyweight) lower-body strength and power outcomes for the GM group (m = 3, f = 5) compared to the AO group (m = 12, f = 5). This result was not seen in the MW group (m = 6, f = 7) compared to the NM group (m = 9, f = 3). Thus, it can be reasonably concluded that the significant relative lower-body strength and power differences seen in our study between the GM group compared to the AO group, and given the similar participant characteristics and even distribution of males and females between groups, potentially repudiates any sex effect. 

Although our testing was comprehensive, we performed a general neuromuscular and physiological testing battery, which did not contain judo-specific fitness testing (e.g., the SJFT). As such, our results may not directly reflect the in-competition nature of judo, but provide efficacy for the use and sensitivity of several field-based tests when assessing junior judokas. Moreover, they can be used routinely as a part of the strength and conditioning component of training. However, we do acknowledge that because of the non-specific nature of these tests, it is difficult to speculate further about other physiological mechanisms that may contribute to the performance outcomes observed. Furthermore, we did not investigate potential differences between upper- and lower-body aerobic power, and therefore, it is unclear whether upper-body aerobic power testing may yield different results. We also acknowledge that, though valid and reliable for adult trained cyclists, the regression equation used in Lamberts et al. [28] and adopted to predict V0_2max_ in the current study may not be reliable for predicting V0_2max_ in junior combat sport athletes. Finally, with respect to the lower-body strength and power outcomes, we did not conduct correlations in our study to other performance variables (e.g., to uchi mata techniques). As such, we can only speculate on how the results presented in our study may impact technical skill proficiency during competition. Future research should seek to expand on our preliminary evidence and explore the correlation(s) between lower-body strength and power and specific throwing technique performance during competition in national, or even international junior judokas. In addition, future research should also aim to explore the reliability of V0_2max_ prediction equations using the Wattbike for junior combat athletes. 

## 5. Conclusions

We have demonstrated that certain physical and physiological qualities may contribute to and/or differentiate gold-medal winning junior national-level judokas from others. Specifically, gold-medal winners displayed significantly greater lower-body maximal strength and peak anaerobic power compared to silver and bronze medalists, and non-medallists. Such information may be important for strength and conditioning professionals to understand the physical characteristics that distinguish gold-medal winning performances within junior national-level judokas. As such, we suggest that professionals working with junior national-level judokas focus on the development of maximal lower-body strength and power. These findings are particularly relevant for junior athletes aiming to compete in national-level competition or above and may also translate to other intermittent combat sports. 

## Figures and Tables

**Figure 1 sports-08-00014-f001:**
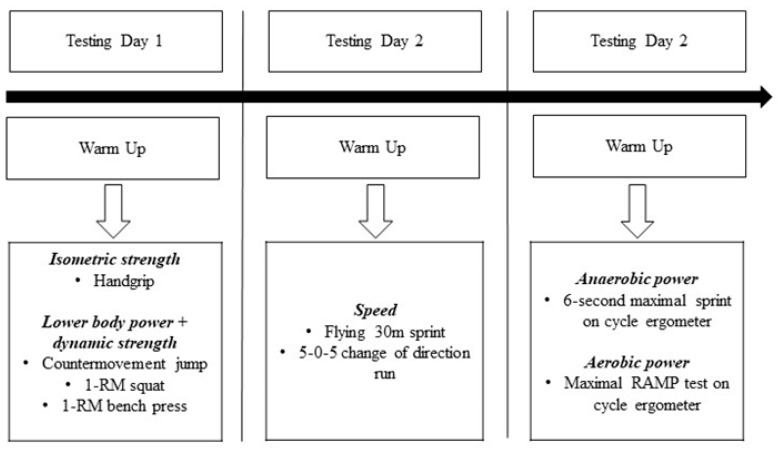
Warm-up and testing timeline, with overview of the testing conducted on each day. BW, bodyweight; m, metre; 1-RM, one-repetition maximum.

**Figure 2 sports-08-00014-f002:**
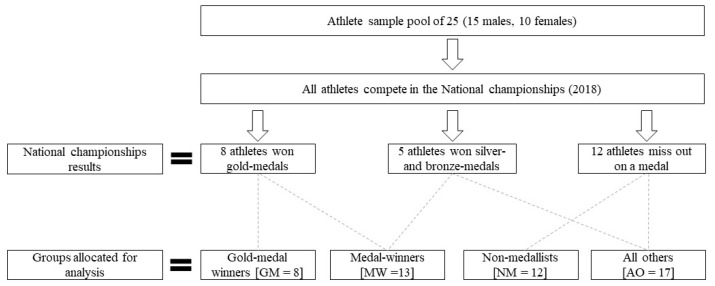
Group allocation schematic. GM, gold-medal winning athletes; MW, medal-winning athletes; NM, non-medallist athletes; AO, all other athletes.

**Figure 3 sports-08-00014-f003:**
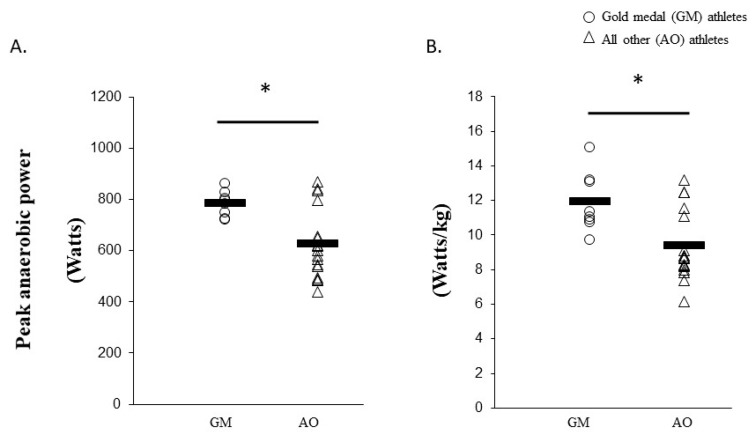
Absolute (watts) (**A**) and relative (watts/kg) (**B**) anaerobic lower-body power results for GM compared to AO. GM, gold-medal winning group; AO, all other athletes. * indicates significant difference (p < 0.05).

**Figure 4 sports-08-00014-f004:**
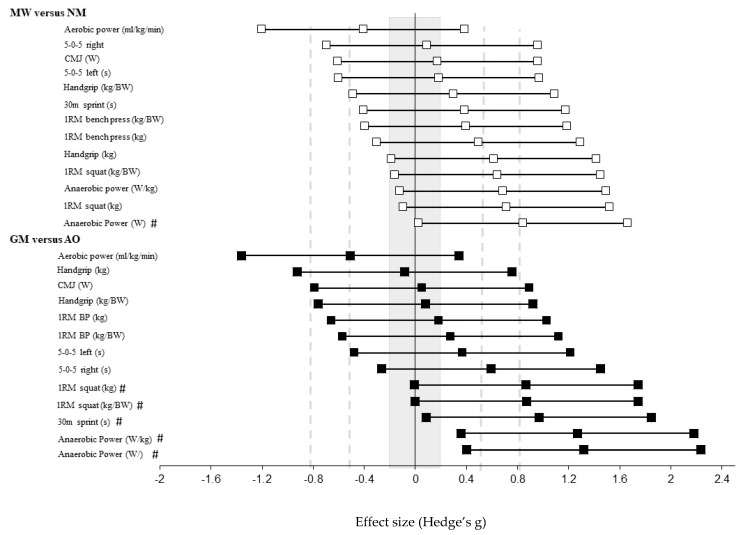
Hedge’s g and 95% confidence interval (CI) of all outcome measures for MW compared with NM, and GM compared with AO. MW, medal winners; NM, non-medallists; GM, gold medal winners; AO, all other athletes. Grey shaded area represents trivial effects size, and grey lines represent cut-off for moderate and large effects, respectively. # indicates large effects.

**Figure 5 sports-08-00014-f005:**
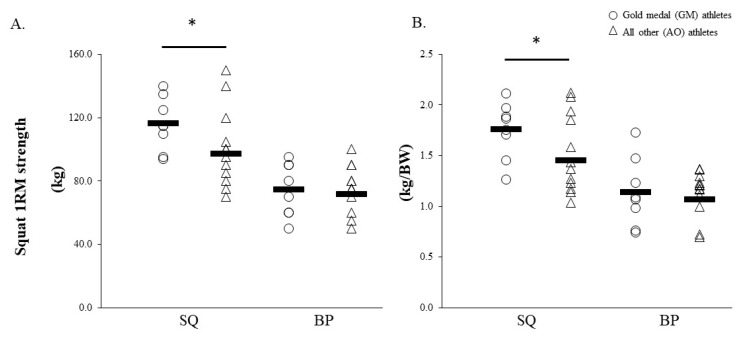
Absolute (kg) (**A**) and relative (kg/BW) (**B**) lower-body strength results for GM compared to AO. GM, gold-medal winning group; AO, all other athletes; SQ, squat; BP, bench press. * indicates significant difference (p < 0.05).

**Figure 6 sports-08-00014-f006:**
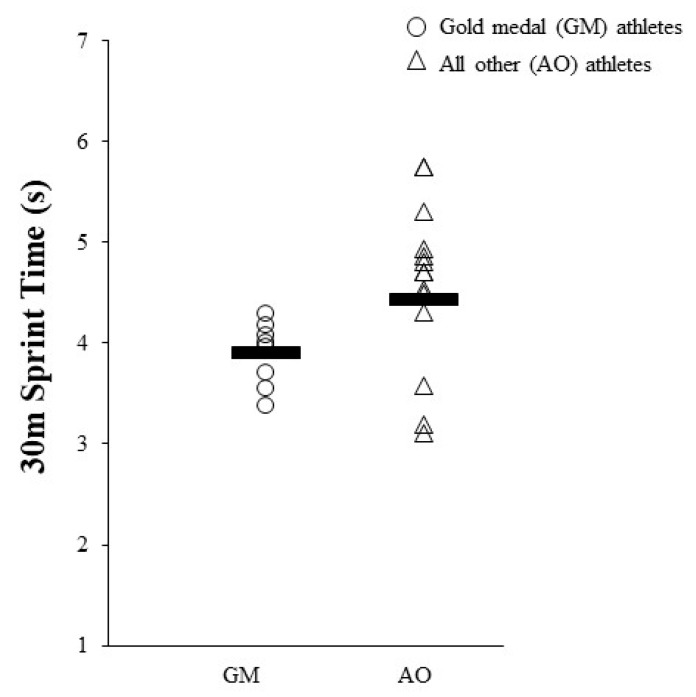
Maximal 30m sprint time for GM compared to AO. GM, gold-medal winning group; AO, all other athletes; s, seconds.

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
