# Peer review of "Absolute and Relative Strength, Power and Physiological Characteristics of Indian Junior National-Level Judokas"

_sports, 2020, doi:10.3390/sports8020014_

Round 1

Reviewer 1 Report

This retrospective study evaluated the potential differences between a variety of physiological performance parameters in junior judokas athletes separated into 4 groups (GM vs. AO and MW vs. NM). It also used a stepwise analysis to predict successful performance. The manuscript is well-written and provides relevant information on how S&C professionals should design certain aspects of their training regimes for these athletes. The study methods are well explained and articulated. I only have a few minor comments for the authors to consider in revising their manuscript:

Line 113 - should be "test-retest" not "test-rest". Line 185 - not sure why there is a -1 after the VO2max? Line 222 - the x-axis on the figure needs a label. Line 253 - on figure 5 should define SQ (Squat) and BP (Bench Press) below. Lines 300-302 - not sure why these lines are here?

Reviewer 2 Report

Harris and colleagues explore physiological characteristics of elite junior judokas in an effort to delineate attributes that predispose these individuals for success. While I found the paper to be well written and the results intuitive, I have the following concerns:

Major Comments

Study design - my largest concern has to do with the construct of the various groups (i.e., GM, MW, NM, AO). Specifically, I find it problematic that many subjects have been binned into multiple groups. Moreover, the "All Others" group seems a bit arbitrary, and this is particularly problematic considering it is the comparisons of this group to the gold medal group that the authors find statistical significance and derive their conclusions. Might the authors be familiar with previous literature to substantiate these groups? Reporting of group characteristics - somewhat related to my first comment, I ask that the authors consider reporting the characteristics of the various groups in a table format. Moreover, while I realize the study may not have been adequately powered to analyze for sex differences, I encourage the authors to construct these tables by sex so that the reader can at least assess the potential for a sex-effect via visual inspection. Aerobic power assessment - the authors employed a RAMP test and then used a regression equation by Lambert et al. to derive oxygen consumption. However, this equation was generated in trained cyclists and thus I have concerns that it may not be applicable to junior athletes. Anaerobic power assessment - please briefly provide rationale for using a 6-s Wingate rather than the traditional 30-s protocol. Calculation of percent difference - this should be computed as: ((Mean Group 1 - Mean Group 2)/Average Group 1&2) * 100 Statistical analyses - Since parameteric testing was employed, were data checked for normality? Results (line 301-302) - please remove

Minor Comments

Abstract (line 27): please fix "analyses were tested between group differences" Introduction (line 75): hypothesis statement should be written in the past tense

Reviewer 3 Report

Dear Authors, it was a great pleasure to review this paper. The research was well designed and the paper is well written in relation to content, style and data presentation. I have only minor suggestions related to participants description and conclusions:

The authors should add "Indian junior...." to the title of the paper. Authors should better describe the length of experience of athletes i.e. how long they generally practice judo? How many medals did they win? If they really have 2-year resistance training, that means they started approx. at the age of 15, how this practice generally looked? Describe in the introduction or methods the level of Indian judo with regard to international competition. "several physical and physiological qualities may contribute" (line 388). It is known without research. The sentence: "Such information may be important for strength and conditioning professionals to understand the physical characteristics that facilitate success within junior national-level judokas" (lines 391-393) is rather too broad. The authors did not explain any mechanisms allowing them to understand the success backgrounds. They found in which characteristics the medal winners are netter than non-medallists athletes.

Round 2

Reviewer 2 Report

Though the authors were able to address many of my minor concerns, I have serious reservations regarding the study design that prohibit me from recommending the paper for publication. In particular, I do not believe the authors have provided significant rationale for the "All Others" group, and I find this to be particularly problematic given that the primary findings of the study are derived from comparisons of this group to the gold medal winners. Furthermore, I specifically requested that the authors "construct tables by sex so that the reader can at least asses the potential for a sex-effect via visual inspection." Though the authors provided a comparison of age, bodyweight and height between young men and women, they failed to provide insight as to the number of me and women in each group. Moreover, these comparisons provide no understanding of potential physiological differences (Table S2) between men and women that are most pertinent. In my opinion, this is a fatal flaw. What if 7 of the 8 "gold medal" winners were men, and more than half of the "all others" group were women? By comparing these groups are we really seeking to determine the physiological pre-requisites for judoka success or are we simply comparing physiological characteristics between young men and women? Unfortunately, in the present form this question cannot be answered.

Round 3

Reviewer 2 Report

I applaud the authors for their persistence and thank them for addressing my comments so quickly. In the present form I believe the manuscript is fit to be accepted.